# Technical note: Quality assessment of ozone reanalysis products and gap-filling over subarctic Europe for vegetation risk mapping

Stefanie Falk[1], Ane V. Vollsnes[2], Aud B. Eriksen[2], Frode Stordal[1], and Terje Koren Berntsen[1]

[1]Department of Geosciences, University of Oslo, Oslo, Norway
[2]Department of Biosciences, University of Oslo, Oslo, Norway

**Correspondence:** Stefanie Falk (stefanie-elfriede.falk@gmx.net)

**Abstract.** We assess the quality of regional and global ozone reanalysis data for vegetation modeling and ozone ($O_3$) risk mapping over subarctic Europe where monitoring is sparse. Reanalysis data can be subject to systematic errors originating from, e.g., quality of assimilated data, distribution and strength of precursor sources, incomprehensive atmospheric chemistry or land–atmosphere exchange, and spatiotemporal resolution. Here, we evaluate two selected global and one regional ozone reanalysis product. Our analysis suggests that global reanalysis products do not reproduce observed ground-level ozone well in the subarctic region. Only the Copernicus Atmosphere Monitoring Service Regional Air Quality (CAMSRAQ) reanalysis ensemble sufficiently captures the observed seasonal cycle. We also compute the root mean square error (RMSE) by season. The RMSE variation between $(2.6 - 6.6)\,\mathrm{ppb}$ suggests inherent challenges even for the best reanalysis product (CAMSRAQ). $O_3$ concentrations in the subarctic region are systematically underestimated by $(2 - 6)\,\mathrm{ppb}$ compared to the ground-level background ozone concentrations derived from observations. Spatial patterns indicate a systematical underestimation of ozone abundance by the global reanalysis products on the west coast of northern Fennoscandia. Furthermore, we explore the suitability of the CAMSRAQ for gap-filling at one site in northern Norway with a long-term record but not belonging to the observational network. We devise a reconstruction method based on Reynolds decomposition and adhere to recommendations by the United Nations Economic Commission for Europe (UNECE) Long Range Transboundary Air Pollution (LRTAP) convention. The thus reconstructed data for two weeks in July 2018 are compared with CAMSRAQ evaluated at the nearest neighboring grid point. Our reconstruction method's performance ($76\,\%$ accuracy) is comparable with CAMSRAQ ($80\,\%$ accuracy) but diurnal extremes are underestimated by both.

## 1 Introduction

Tropospheric ozone ($O_3$) as a secondary pollutant is highly toxic and harmful to human health (WHO - World Health Organization, 2008; Fleming et al., 2018) and a variety of ecosystems globally (Mills et al., 2011, 2018; Emberson, 2020). At the same time $O_3$ acts as a potent greenhouse gas (Myhre et al., 2013). Ozone causes an estimated annual global yield loss of four major crops (wheat, rice, maize, and soybean) of $3 - 15\,\%$ (Ainsworth, 2017) and threatens food security in rapidly developing countries, e.g., in East and South-East Asia (Tang et al., 2013; Tai et al., 2014; Chuwah et al., 2015; Mills et al., 2018).

In the troposphere, $O_3$ is produced in complex chemical cycles involving precursors such as carbon monoxide (CO) and hydrocarbons (e.g. methane, terpenes) in the presence of nitrogen oxides ($NO_x$) (Monks et al., 2015). These hydrocarbons can be of anthropogenic or natural origin and are often referred to as volatile organic compounds (VOCs) and biogenic VOCs (BVOCs), respectively. The primary sink of $O_3$ in the troposphere is dry deposition to different surfaces of which the removal by vegetation amounts to over $50\%$ (Monks et al., 2015; Clifton et al., 2020). Plants take up $O_3$ through their stomata (leaf openings for gas exchange). In the leaf interior, $O_3$ induced radical oxygen species (ROS) damage cell membranes leading to necrosis and ultimately to programmed cell death (Kangaskärvi et al., 2005). Ozone damage is considered to accumulate over time. To assess the potential risk posed by ozone, various metrics have been defined. Mills et al. (2011) showed that the Phytotoxic Ozone Dose over a threshold y ($POD_y$) (integrated flux through the stomata) is capable of capturing observed negative effects on crops and semi-natural vegetation (e.g. clover) better than an integrated exceedance over a fixed concentration threshold (e.g. $40\,ppb$). Furthermore, $O_3$ uptake and subsequent damage negatively affect photosynthesis and stomatal conductance (e.g., Pellegrini et al., 2011; Watanabe et al., 2014). This, in turn, reduces gross primary production (GPP) (Lombardozzi et al., 2015b, a; Hoshika et al., 2015) and has the potential to off-set growth effects of carbon dioxide ($CO_2$) fertilization in the future (Franz and Zaehle, 2021) as well as to induce measurable positive feedback on surface temperatures in highly polluted regions (Zhu et al., 2021).

Due to these risks, $O_3$ is included in air quality monitoring networks under the WMO (World Meteorological Organization) Global Atmosphere Watch (GAW) program. Remote regions in the Arctic and subarctic, however, are scarcely covered (refer to Section 2 for the coverage of northern Fennoscandia). With climate change already promoting an earlier and longer growing season (Linderholm, 2006; Karlsen et al., 2007; Høgda et al., 2013), subarctic vegetation may become more vulnerable to damage induced by cumulative $O_3$ uptake in the future. Although, species acclimated to the Arctic and subarctic climates were not found to be more sensitive to ozone than species in less extreme environments (Karlsson et al., 2021).

$O_3$ as well as its precursors is subject to atmospheric transport causing pollution peaks in the otherwise pristine Arctic and subarctic environments (Stevenson et al., 2005; Young et al., 2013). This long-range transport of pollutants has been identified as one of the main sources of enhanced $O_3$ concentration ($[O_3]$) in Fennoscandia (Andersson et al., 2017). Here, $[O_3]$ refers to the concentration as volume mixing ratio (VMR) of ozone in $ppb$. Peak $[O_3]$ in summer is often a combination of stagnant weather situations accompanied by heatwaves and enhanced precursor emissions due to extensive forest fires (e.g. in 2003, 2006, 2018) (Lindskog et al., 2007; Karlsson et al., 2013). The prominently elevated $[O_3]$ which occurs in April/May over northern Fennoscandia is caused by other factors. This so-called ozone spring peak can be attributed to a build-up of $O_3$ and precursors due to a suppression of removal from the troposphere during the polar night and their photo-chemical reactivation come spring (Monks, 2000). Tropopause folding events are another contributor and cause an intrusion of dry and $O_3$ rich air masses from the stratosphere (Škerlak et al., 2015).

As indicated above, $POD_y$ is an integrated $O_3$-flux quantity. A proper assessment of $POD_y$ relies on a set of complete, 1-hourly meteorological and ozone data. Since gaps in observational data are common, many techniques of varying complexity have been devised for filling these. The applicability often depends on the shape of the variables' signal, e.g. prominence of the diurnal cycle. In the simplest case of monotonously increasing/decreasing data and little fluctuation, a first-order polynomial

may suffice. In the following, we give an account of the detailed practical recommendations by Mills et al. (2020). For gaps of less than $5\,\mathrm{h}$, gap-filling with an average value over the preceding and subsequent time steps is recommended. This method, however, does not suffice for observables such as $O_3$ that display a distinct diurnal cycle and leads to an underestimation around noon and an overestimation during the night, respectively. Similarly, gaps longer than $5\,\mathrm{h}$ but less than $24\,\mathrm{h}$ ought to be filled by averaging the preceding and subsequent day at each time step. For gaps exceeding $1\,\mathrm{d}$, Mills et al. (2020) suggest exploiting data from close-by monitoring stations with a Pearson correlation coefficient $r^2$ of preferably $0.6$ or higher. A period of at least one season ($3\,\mathrm{months}$) is recommended for this statistical analysis. To account for the seasonal variability, the projection between sites is to be computed for the same season the gap occurred. Where available, auxiliary data from model reanalysis can be used.

As indicated above, reanalysis data can be used for gap-filling, but more often it is used to study emerging trends in tropospheric ozone in remote regions such as the Arctic and subarctic, where scarce observations have to be supplemented with model simulations. Atmospheric reanalyses are based on a fixed state of an operational data assimilation system used for forecasts ingested with the most complete set of observational data. In terms of atmospheric chemistry, this includes meteorological data as well as observations of chemical substances from, e.g., satellite, airborne instruments, and ground-level monitoring station networks. Global reanalyses, however, have already been shown to underestimate $[O_3]$ particularly over the polar region (Huijnen et al., 2020; Barten et al., 2020). Barten et al. (2020) suggest, that global reanalysis products that only assimilate satellite products do not sufficiently cover $[O_3]$ variations. The large discrepancies can be explained by the low spatiotemporal resolution not capturing atmospheric boundary layer dynamics and missing processes such as mechanistic ocean–atmosphere $O_3$ exchange.

In the following, we evaluate and validate the quality of three reanalysis products concerning surface ozone over northern Fennoscandia with available long-term observations. All data are presented in Section 2. In Section 3, we derive a generalized ozone climatology for northern Fennoscandia from in situ observations and quantify the overall quality of the ozone reanalysis. We look at the respective seasonal cycles, spatial patterns, and derive the relative impact on an integrated-flux metric. Based on these results, we provide a methodology for reconstructing missing observational data over an extended period of several weeks based on Reynolds decomposition and compare it with the evaluation of the best reanalysis product at the nearest neighboring grid point. We close with discussions and conclusions (Section 4).

## 2 Data

In this section, we present long-term ground-level $O_3$ observation data for our focus area, northern Fennoscandia, which we define here as north of $67.5\,^{\circ}\mathrm{N}$, and determine their correlation. To this end, we compute Pearson correlation coefficients pair-wise. All observational data are taken from the EBAS atmospheric database operated by the Norwegian Institute for Air Research (NILU). We also present the selected ozone reanalysis products provided by the European Centre for Medium-Range Weather Forecasts (ECMWF) and the Copernicus Atmospheric Monitoring Service (CAMS).

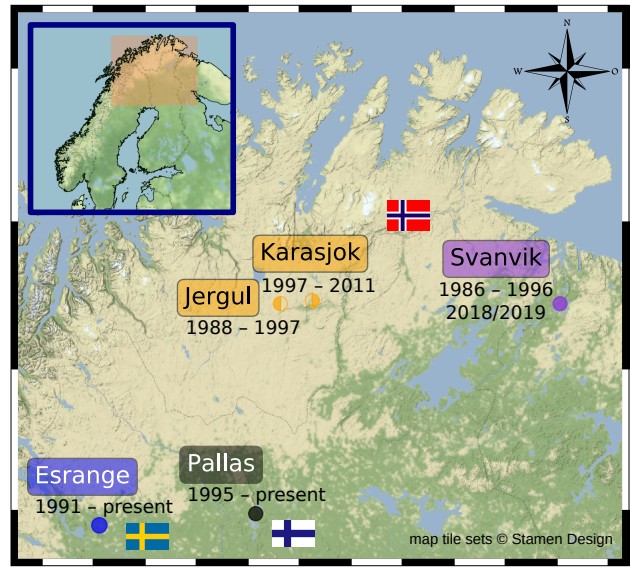

**Figure 1.** Subarctic Europe north of $67.5\,^{\circ}$N, here referred to as northern Fennoscandia. Locations of past and present ozone observation sites used in this study. For more details see Table 1. The introduced color coding for the monitoring sites is used throughout.

## 2.1 Ozone monitoring sites

Northern Fennoscandia is sparsely covered by sites that monitor ground-level background $[O_3]$ and report to the EBAS atmospheric database (Fig. 1). A detailed overview over the past and present ozone monitoring sites in northern Fennoscandia with a considerable duration of data acquisition is given in Table 1. Continuous ozone data are available as early as mid-1986 from the NILU atmospheric monitoring site at Svanvik located in the Pasvik valley. Measurements, however, did not continue after 1996. To supplement field experiments on subarctic vegetation, we installed an ozone monitor at Svanvik exclusively for the growing seasons 2018/19 in collaboration with NILU. Due to irregularities in data acquisition, two weeks of data are missing from the record in July 2018. These shall be subject to our proposed data reconstruction (Section 3.3). At the same latitude but further west, a station was established in the early 1990s above the Karasjohka river valley. Originally placed at Jergul the station was later moved downstream closer to the city of Karasjok using the same equipment but increasing the recorded floating-point precision of the ozone monitor. The station was decommissioned in 2011. Data series from Svanvik and Jergul are highly uncertain because of insufficient quality control and irregular calibration before 1997 which led to degradation of the monitors over time and introduced drifts in the ozone data series (Solberg, 2003). Solberg (2003) further reported a systematic uncertainty for these data of the order of $10\,\%$, which they deemed too large to conduct a strict trend analysis of ground-level background $[O_3]$. For our purpose of evaluating seasonal cycles on a climatological timescale, we can consider these uncertainties as small enough. Further south, two stations have been established at Esrange (Sweden) and Pallas (Finland) in the early 1990s. Data are available from EBAS until the end of 2018 and 2019, respectively (last accessed April 2021).

**Table 1.** Past and present ozone observation sites in northern Fennoscandia. Data available from EBAS.

| Name | Country | ID | Location | | | Operational |
|------|---------|-----|----------|----------|----------|-------------|
| | | | lat | lon | alt | |
| | | | (°N) | (°E) | (m) | |
| Esrange | SWE | SE0013R | 67.83 | 21.07 | 475 | $1991-2018^{\dagger}$ |
| Jergul | NOR | NO0030R | 69.45 | 24.60 | 255 | $1997-2011$ |
| Karasjok | NOR | NO0055R | 69.467 | 25.217 | 333 | $1988-1997$ |
| Pallas | FIN | FI0096G | 67.97 | 24.12 | 565 | $1995-2019^{\dagger}$ |
| Svanvik | NOR | NO0047R | 69.45 | 30.03 | 30 | $1986-1996^{\ddagger}$ |

$^{\dagger}$ Data availability on EBAS at present.

$^{\ddagger}$ Exclusive monitoring in growing season 2018/19.

In Fig. 2, daily mean ozone concentration climatologies ($\langle[O_3]\rangle$) for the data taken at Esrange, Jergul/Karasjok, Pallas, and Svanvik are shown together with their respective standard error ($\sigma_{\langle[O_3]\rangle} = \frac{\sigma_{[O_3]}}{\sqrt{n}}$). The annual average $\langle[O_3]\rangle$ at Svanvik is $6.6\,\mathrm{ppb}$ lower compared to the other sites. This can be attributed to the station's location at lower altitude and amidst agriculturally used land surrounded by forests in contrast to Pallas where the vegetation consists of low vascular plants, mosses, and lichen (Hatakka et al., 2003). An increase in ground-level background $[O_3]$ since the early 1990s cannot be dismissed. Given 2019 was a climatologically normal year, we estimate the deviation from the 1990s ozone climatology at Svanvik $\delta[O_3] = (1.2 \pm 5.0)\,\mathrm{ppb}$. The $\delta[O_3]$ indicates a small and statistically insignificant increase in $[O_3]$.

The Pearson correlation coefficients ($r^2$) for the combined data set of Jergul/Karasjok show a high correlation with Esrange ($r^2 = 0.78$) as well as Pallas ($r^2 = 0.79$). We, therefore, combine observational data from Esrange, Jergul/Karasjok, and Pallas to derive a generalized ozone climatology for northern Fennoscandia which represents the expected ground-level background in this region. The correlation of Svanvik with Esrange is fair ($r^2 = 0.42$), but good with Pallas ($r^2 = 0.61$). The climatologies displayed in Fig. 2 cover the known features of the ozone seasonal cycle in northern Fennoscandia well and reflect the expected increase of ozone abundance with altitude where Pallas is located at the highest altitude and Svanvik at the lowest. The highest average ozone concentration ($\langle[O_3]\rangle_{\max} = (46.35 \pm 0.17)\,\mathrm{ppb}$) is regularly observed in April/May and the lowest average concentration is reached in August/September ($\langle[O_3]\rangle_{\min} = (24.18 \pm 0.18)\,\mathrm{ppb}$). The $\sigma_{\langle[O_3]\rangle}$ lie well below $0.5\,\mathrm{ppb}$ for Esrange, Jergul/Karasjok, and Pallas. This is considerably lower than at Svanvik ($0.3\,\mathrm{ppb} < \sigma_{\langle[O_3]\rangle} \leq 1\,\mathrm{ppb}$) and can be attributed to the length of these time series, a better quality control, and less diurnal variability at higher altitudes.

## 2.2 Ozone reanalysis

From the global reanalysis products available from ECMWF that include atmospheric tracers, including ozone, we select the Monitoring Atmospheric Composition and Climate (MACC) and the latest Copernicus Atmosphere Monitoring Service reanalysis (CAMSRA) (Inness et al., 2013, 2019). The temporal, as well as spatial resolution of these reanalysis products is rather coarse: 3-hourly and $0.75° \times 0.75°$ or roughly $29.3\,\mathrm{km} \times 83.4\,\mathrm{km}$ at the location of Svanvik. From the Copernicus

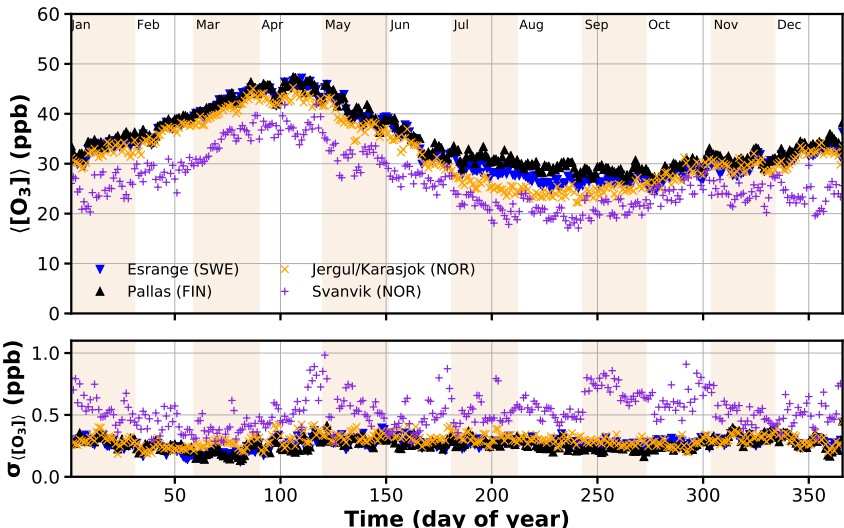

**Figure 2.** Daily mean ozone climatologies (upper panel) and standard error (lower panel) over the day of the year. All stations located in northern Fennoscandia with data records exceeding $10\,\text{years}$ are displayed. The data taken at Jergul and Karasjok have been combined.

Atmospheric Monitoring Service Regional Air Quality (CAMSRAQ) system, surface ozone reanalysis ensemble means are available for a European domain. The CAMSRAQ is based on nine European state-of-the-art numerical air quality models. The ensemble mean is at higher spatial and temporal resolution compared to the global reanalyses: $0.1° \times 0.1°$ (roughly $3.9\,\text{km} \times 11.1\,\text{km}$ at Svanvik) and 1-hourly. The periods covered differ but no data is available before the turn of the millennium. The CAMSRA is available in near real-time and covers a period of sufficient length for climate analysis (2003 – present). For this study, a shorter subset of CAMSRA (2003–2012) has been chosen for comparability with the MACC reanalysis in terms of statistical uncertainties. Predominately, the CAMSRAQ system is used for air quality forecasting and the reanalysis has currently not been extended beyond 2018.

All reanalysis products, apply the latest version of the operational weather forecast system (OpenIFS) of ECMWF to force their models. Concerning the assimilated observational ozone data, all reanalysis products differ. The MACC reanalysis assimilates only satellite-derived tropospheric column ozone, while CAMSRA also includes ozone profiles from satellite retrievals. In situ observations from ozone near-surface station networks are only assimilated in the CAMSRAQ reanalysis ensemble. All relevant details concerning the reanalysis data sets are listed in Table 2.

The MACC reanalysis is still well known and used in the wider community, albeit its lower accuracy compared to CAMSRA (Huijnen et al., 2020). To assess whether and how the improvements to the CAMS assimilation system affect the reanalysis results in our focus area, we analyze both MACC and CAMSRA. CAMSRAQ has been specifically chosen to test whether a higher spatiotemporal resolution will also give better results in our focus area.

On global scales, at least two other ozone reanalysis products are available, the Tropospheric Chemistry Reanalysis (TRC) 1 and 2 (Miyazaki et al., 2020) and the Japanese Reanalysis 55 (JRA-55) (Kobayashi et al., 2015). As part of the comprehensive

**Table 2.** Global/regional ozone reanalysis products used in this study.

| Name | Provider | Resolution | | | Time period | Meteorological forcing | $O_3$ assimilation |
|------|----------|---------|----------|----------|-------------|------------------------|-----------------|
| | | spatial | temporal | vertical | | | |
| MACC | ECMWF | $0.75° \times 0.75°$ | 3-hourly | $10\,m^{\triangleright}$ | 2003 – 2012 | OPS | satellite $^{\triangledown}$ |
| CAMSRA | ECMWF | $0.75° \times 0.75°$ | 3-hourly | $10\,m^{\triangleright}$ | 2003 – 2012 $^{\dagger}$ | ERA5 / OPS $^{\ddagger}$ | satellite $^{\blacktriangledown}$ |
| CAMSRAQ | Copernicus | $0.1° \times 0.1°$ | 1-hourly | surface | 2014 – 2018 $^{\dagger}$ | OPS $^{\star}$ | in situ $^{\triangle}$ |

$^{\triangleright}$ Layer thinckness at ground level, same as for operational IFS; $^{\dagger}$ Subset of reanalysis data used in this study. $^{\ddagger}$ ERA5 (2003–2016), OPS (later); $^{\star}$ EURAD uses WRF for downscaling of operational IFS; $^{\triangledown}$ MLS, OMI - tropospheric column; $^{\blacktriangledown}$ SCIAMACHY, MIPAS, MLS, OMI, GOME2, SBUV2 - tropospheric column + profile; $^{\triangle}$ METEO France NRT.

reanalysis inter-comparison study by Huijnen et al. (2020), TRC-1 and 2, CAMS interim reanalysis, and CAMSRA were by means of seasonal averages. The results suggested a similar performance of CAMSRA and TRC-2 in our focus area. Therefore, we assume our selection to be representative of the state-of-the-art global reanalysis products.

The comprehensive JRA-55 reanalysis is the longest reanalysis dataset available spanning several decades. With a horizontal resolution of T319, 6-hourly temporal resolution, and interpolation to pressure levels (e.g. $1000\,hPa$) it is too coarse and not suitable for our purpose.

## 3  Analysis

In the following, we assess the quality of the reanalysis products, MACC, CAMSRA, and CAMSRAQ, with respect to the generalized ozone climatology derived from ground-level ozone observations in northern Fennoscandia. We focus in particular on the seasonal cycle of $[O_3]$ with its prominent peak in spring and dip in late summer and identify the reanalysis product that best reproduces these features. Concerning ozone risk mapping, we assess implications on an integrated-flux metric that is similar to $POD_y$. We then devise a reconstruction method for missing data applicable for extended periods of data gaps based on Reynolds decomposition and compare with the best reanalysis product evaluated at the nearest neighboring grid point of Svanvik.

### 3.1  Quality of ozone reanalysis products in northern Fennoscandia

First, we evaluate the reanalysis products qualitatively at the site level. We compare the seasonal cycle of the generalized ozone climatology with seasonal cycles derived for each reanalysis product at the nearest neighboring grid point of the actual monitoring sites. In this way, we can also test the vertical resolution of the products concerning the expected ozone abundance in response to differing ground-level altitudes.

The generalized ozone climatology and its respective standard deviation (gray band) shown in Fig. 3 are based on a spline fitted through the climatological daily mean $[O_3]$. The global products (MACC, CAMSRA) do not reproduce the observed seasonality of ground-level $[O_3]$ well. The MACC reanalysis (Fig. 3a) reveals a strong negative deviation (bias) amounting to

$-(9 \pm 7)$ ppb on average and displays no distinct seasonal cycle. The ozone climatology is rather flat throughout the whole cycle with a small peak in March. MACC $[O_3]$ is considerably too low compared to the generalized climatology in all seasons but summer. The March peak is followed by a flattening and a second peak in July. The seasonal low is shifted towards November/December.

CAMSRA matches the observed ozone climatology poorly (Fig. 3b). Despite reproducing $[O_3]$ well during the growing season (May–October), it does not reproduce the actual seasonality in northern Fennoscandia. The CAMSRA derived spring peak lags behind observations by 1 month and is 5 ppb too low, whereas the minimum occurs in January compared to August/September. In general, CAMSRA fails in reproducing $[O_3]$ in all seasons but summer. The annual amplitude $((26\pm1)$ ppb) is larger than in the climatology derived from observations (19 ppb). Both global reanalysis products place the $O_3$ abundance evaluated at the location of Svanvik highest. This indicates an insufficient vertical resolution of these models. This is important in terms of usage for gap-filling as well as Europe-wide or global risk assessment concerning the Arctic and subarctic vegetation that may rely on these data.

In contrast, the ensemble mean of the CAMSRAQ reproduces the seasonal cycle in northern Fennoscandia well (Fig. 3c). CAMSRAQ correctly depicts $[O_3]$ at Svanvik lower than at the other sites most likely due to the higher resolution and data assimilation of in situ ozone observation. On average, CAMSRAQ slightly underestimates $[O_3]$ $(-(2.8 \pm 0.5)$ ppb) compared to observations.

However, the reanalysis products' time series are not sufficiently long enough to study deviations from the observed climatology with a high statistical significance. Though, the associated standard deviation is usually smaller in models compared to observations due to the inherent spatiotemporal averaging. This has no impact on our qualitative results. Recent analyses indicate a levelling or decline of tropospheric background $[O_3]$ over Europe after 2007 (Cooper et al., 2014; Wespes et al., 2018; Gaudel et al., 2018) following a steady increase over the past decades (IPCC - Intergovernmental Panel on Climate Change, 2013, Chapter 2). This indicates that the observation-based generalized northern Fennoscandia climatology which includes data before 2007 could be biased towards a higher annual average $[O_3]$. As estimated in Section 2.1, the climatology derived for Svanvik is insignificantly underestimating present day $[O_3]$.

In Fig. 4, the seasonally averaged deviation is shown between each reanalysis product and the generalized ozone climatology which shall represent the expected ground-level ozone background for the whole region. We also computed the root mean square error (RMSE) over land-only which is displayed in the upper left corner of the respective panel. As expected, the global reanalysis products, MACC and CAMSRA (Fig. 4a,b), show substantial negative deviations $(\Delta[O_3] < -10$ ppb) in winter (DJF) and spring (MAM). The respective RSMEs range between $(12.3 - 15.2)$ ppb (MACC) and $(10.1 - 15.6)$ ppb (CAMSRA). The smallest deviations $(\Delta[O_3] > -4$ ppb) occur in summer (JJA). In Summer, the MACC reanalysis deviations are overall negative except for a small region east of Tromsø where $\Delta[O_3]$ are slightly positive (RSME = 3.9 ppb). While a positive $[O_3]$ deviation would be expected over the Scandinavian Mountains due to the higher elevation compared to the reference height of the generalized climatology, the spatial pattern of the MACC reanalysis displays lower $[O_3]$ in coastal areas in the west which could point to an influence of oceanic fractions in these grid cells. The lowest deviations occur in areas with mean elevations similar to the generalized climatology. Especially in Summer, CAMSRA shows a distinctive gradient

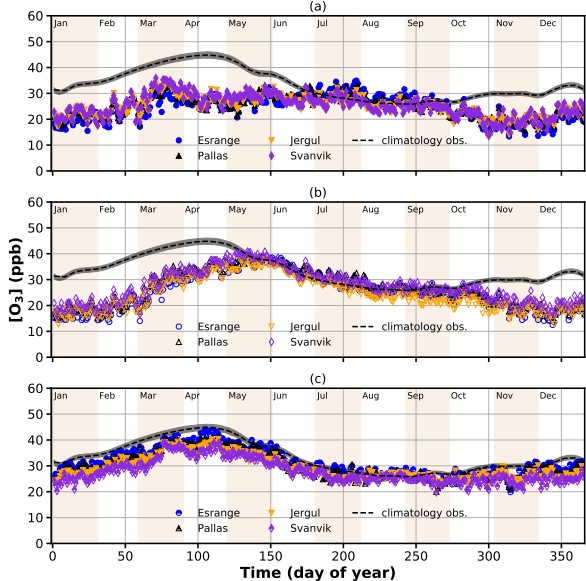

**Figure 3.** Daily mean ozone climatologies computed from the ozone reanalysis products (a) MACC; (b) CAMSRA; (c) CAMSRAQ ensemble mean. The reanalysis products were evaluated at the nearest neighboring grid point of the featured monitoring sites to assess also the vertical resolution. The generalized ozone climatology shown as a gray band represents the expected seasonal cycle of ground-level ozone background $O_3$ in northern Fennoscandia. On average, all reanalysis products underestimate $[O_3]$.

with positive deviation furthest East, in areas surrounding the northern Gulf of Boothia. Similar to MACC, coastal areas in the west seem to be influenced by oceanic fractions in these grid cells. The deviation of CAMSRAQ from the generalized ozone climatology is considerably smaller than for the global reanalysis products and stays below $20\%$ ($\mathrm{RSME} \leq 6.6\,\mathrm{ppb}$) at all times. The white areas at the northern and eastern borders represent the domain borders (Fig. 4c). The largest deviations are again found in winter and spring, while the smallest occur in summer ($\mathrm{RSME} \leq 2.6\,\mathrm{ppb}$)). The deviation in ozone follows the terrain more closely. Consistent with the on average too low ozone abundance, the highest negative deviations are displayed in areas that lie at a lower elevation than the reference stations of the generalized climatology.

The performance of CAMSRAQ ensemble and each of its contributing models is continuously validated with data from active European monitoring stations south of $60.53\,°\mathrm{N}$. This validation is graphically provided on the Copernicus website (last accessed in May 2021). Following the Copernicus Atmosphere Monitoring Service (2020) guidelines, the analysis comprises mean bias, modified mean bias, RMSE, fractional gross error, and temporal correlations of the $O_3$ daily maximum. The ensemble median of the $O_3$ daily maximum shows the largest RMSE in JJA ($5.28\,\mathrm{ppb}$) and the smallest in MAM ($4.05\,\mathrm{ppb}$) which is contrary to our results for the daily mean $O_3$. The mean bias oscillates between $0.97\,\mathrm{ppb}$ (DJF) and $-1.77\,\mathrm{ppb}$ (JJA) which is also opposite to our evaluation in northern Fennoscandia with a small bias in JJA and a larger negative deviation from observations in DJF and MAM. This indicates that underlying uncertainties in CAMSRAQ manifest differently at higher

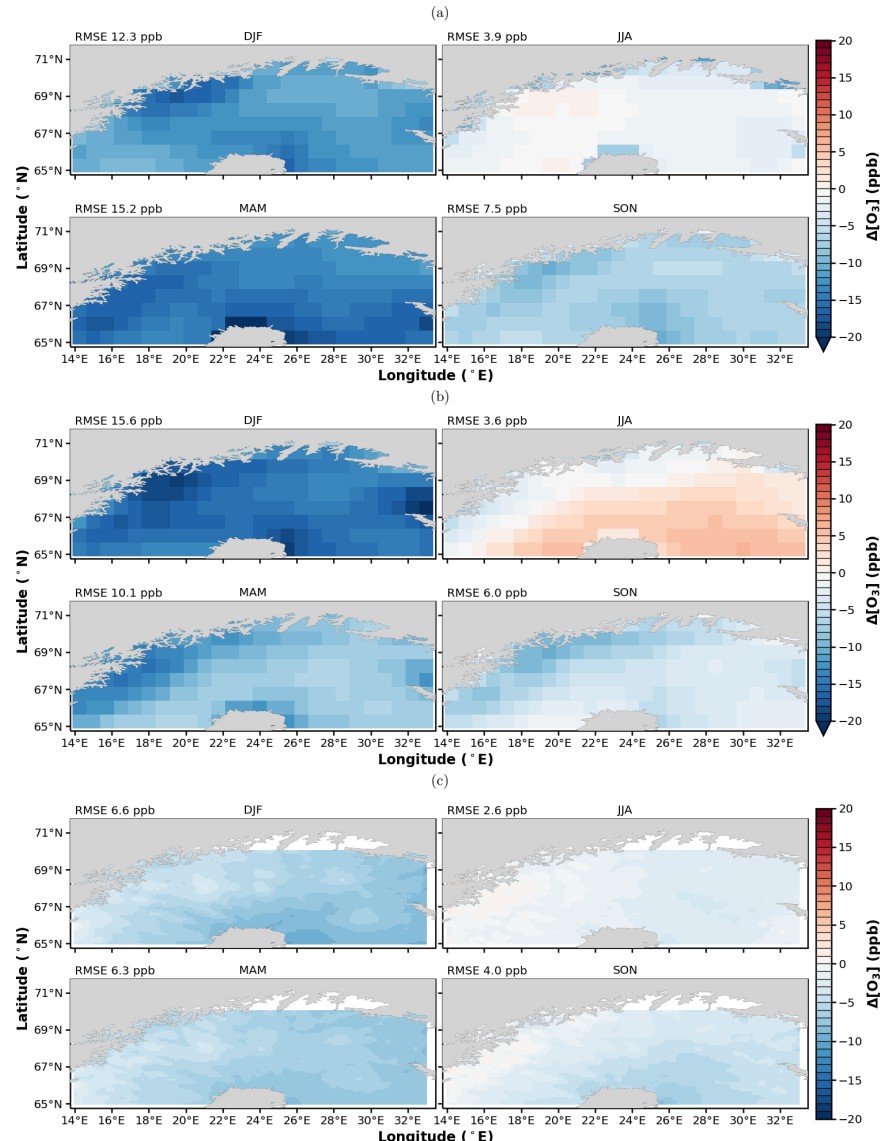

**Figure 4.** deviation of reanalysis products from generalized ozone climatology for northern Fennoscandia: (a) MACC; (b) CAMS; (c) CAMSRAQ. Negative/positive values indicate that the reanalysis product underestimates/overestimates the ground-level background $[O_3]$. Shown are seasonal averages: December/January/February (DJF); June/July/August (JJA); March/April/May (MAM); September/October/November (SON). The RMSE has been computed over land-only and is displayed in the upper left corner of each panel.

latitudes. Enhancements that lead to better model performances in mid latitudes, hence, do not necessarily affect results in the Arctic and subarctic in the same way.

## 3.2 Implications on integrated flux quantities

As pointed out by Hayes et al. (2018), the highest sensitivity to differences in ozone concentrations occurs in coincidence with the highest productivity of plants in summer. The poor agreement between the global reanalysis products and observations in winter and fall may therefore have limited consequences on integrated flux quantities (e.g. $POD_y$) used to assess the ozone risk on vegetation.

The computation of $POD_y$ is non-trivial and depends on state functions of the atmosphere, soil, and vegetation, as well as wind fields (Mills et al., 2017). In the following assessment, we, therefore, make some simplifications. We choose Svanvik as an example location for which we have meteorological conditions readily available and compute a Cumulative Uptake of Ozone (CUO) with a threshold of $y = 0$

$$CUO_0 = \sum_i \Phi^{O_3}(t_i) \cdot \Delta t, \tag{1}$$

with $\Delta t = 1\,\mathrm{h} = 60^2\,\mathrm{s}$. The time dependent ozone flux through the stomata is usually defined as

$$\Phi^{O_3}(t_i) = [O_3](t_i) \cdot g_{sto}(t_i) \cdot \frac{r_c}{r_c + r_b}. \tag{2}$$

We neglect the quasi laminar ($r_b$) and leaf surface resistance ($r_c$) terms in the following. This can be justified by only looking at the relative percentage differences in the following and not the absolute CUO values. Ozone concentrations are converted from ppb to $\mathrm{mmol\,m^{-3}}$ by using the ideal gas law ($V^{-1} = \frac{P}{RT}$) and multiplying with $10^{-6}$. For simplicity, we assume standard pressure ($P_{std} = 1.013 \cdot 10^{-5}\,\mathrm{Pa}$) but insert observed 2018 temperatures at Svanvik. The stomatal conductance follows from (Jarvis, 1976; Emberson et al., 2000; Mills et al., 2017):

$$g_{sto} = g_{max} \cdot f_{light} \cdot \max\{f_{min}, f_T \cdot f_{VPD} \cdot f_{SWP}\}, \tag{3}$$

with normalized response functions to light ($f_{light}$), temperature ($f_T$), vapor pressure deficit ($f_{VPD}$), and soil water potential ($f_{SWP}$), as well as the minimum ($f_{min}$) and maximum conductance ($g_{max}$). We assume a sufficiently moist soil and hence the dependency on soil water potential to be negligible ($f_{SWP} = 1$).

Meteorological data (temperature, relative humidity, global irradiance) from Svanvik in 2018 is used to compute $g_{sto}$. Although 2018 was characterized by an extended drought period over large parts of Europe, northern Fennoscandia was affected to a lesser degree than the rest of Europe (Gangstø Skaland et al., 2019). We calculate CUO for parameterizations of boreal deciduous and coniferous trees (Table III.11 in Mills et al., 2017). We use the bias-corrected observed ozone climatology for Svanvik as a reference to probe the climatologies based on MACC, CAMSRA, and CAMSRAQ (Fig. 3). For this purpose, MACC and CAMSRA climatologies have been upsampled to 1-hourly resolution by linearly interpolating between existing values.

We find that all reanalysis products overestimate CUO compared to observations (Table 3). CAMSRAQ performs best displaying only a small deviation (2 %). While CAMSRA represented the seasonal cycle better than MACC (Section 3.1), its

**Table 3.** Relative percentage difference in $CUO_0$ for the different ozone reanalysis products compared to observed ozone. Boreal parameterizations of deciduous and coniferous trees are taken from Mills et al. (2017).

| Name | Species | |
|------|---------|---|
| | deciduous | coniferous |
| MACC | 8.1 | 6.4 |
| CAMSRA | 17.15 | 17.12 |
| CAMSRAQ | 2.0 | 1.9 |

performance in terms of CUO is poor. This can be attributed to a pronounced bias towards higher ozone concentrations in CAMSRA during summer as emerges clearly from Fig. 4. The deficits of MACC in spring reduces CUO for coniferous trees and thus counters too high $[O_3]$ in summer.

### 3.3 Reconstruction of missing ozone data

Based on our assessment, only the CAMSRAQ product suffices for gap-filling. We shall now derive a reconstruction method based on a Reynolds decomposition for use in ozone impact studies on vegetation. We will compare the reconstructed data with an evaluation of CAMSRAQ at the nearest neighboring grid point and compute the respective RSMEs with respect to observed data before and after the gap.

The ozone data was taken at Svanvik in 2018. Due to problems in data acquisition, 9–23 July 2018 are missing from the record. These coincide with large, active forest fires in central Sweden (Björklund et al., 2019) which presumably caused elevated concentrations of ozone precursors. Enhanced $[O_3]$ were observed throughout July and coincident peak concentrations above $40\,\mathrm{ppb}$ are found in the data series from Esrange and Pallas on July 4, 12–16, 25, and 31 (Fig. 6a). At Svanvik, the peak $[O_3]$ in early July was not observed but elevated $[O_3]$ occurred at the end of the month. During these forest fire-induced pollution events, $[O_3]$ deviated from the respective climatology by up to $28\,\mathrm{ppb}$ (Fig. 6b). These special conditions demand a more elaborate gap-filling procedure than suggested by Mills et al. (2020). As described in Section 1, gap-filling is usually done by using mean values from the same period from previous years or by using mean values from the same time of day from previous days. Considering forest fires are rare events, those mean values will not be good candidates for gap-filling. In addition, data from a reference station selected based on a high correlation factor alone is not sufficient, because a correlation does not account for systematic offsets or the transport of pollutants.

A Reynolds decomposition is an analytical method often used in atmospheric and climate science to separate the expected value ($\bar{u}$) of a variable $u$ from its fluctuations ($u'$):

$$u = \bar{u} + u'. \tag{4}$$

As expected value, we assume the averaged seasonal cycle from a subset of ozone monitoring data excluding the year of interest and refer to this as ozone climatology $\langle [O_3] \rangle$. The fluctuations $\Delta[O_3]$ (anomalies) for the year of interest are derived

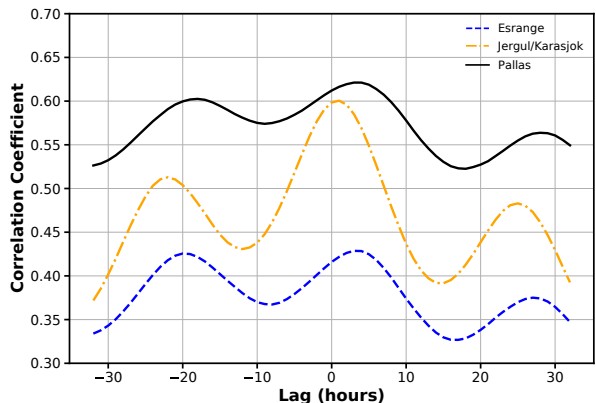

**Figure 5.** Temporal correlation of $[O_3]$ data between Svanvik and other ozone monitoring sites in northern Fennoscandia over time lag. The time lag correlation has been computed by shifting one of the series by $\Delta t$. A negative lag means that Svanvik lags behind, while a positive lag mean the other station lags behind. The highest correlation with Pallas/Esrange is found at a time lag of $3\,\mathrm{h}$, for Jergul/Karasjok at $1\,\mathrm{h}$.

in accordance to Eq. (4):

$$\Delta[O_3] = \langle[O_3]\rangle - [O_3]. \tag{5}$$

To synchronize the time series temporally, we compute time-lagged correlations between Svanvik and the other stations in northern Fennoscandia during the overlapping periods in the 1990s (Fig. 5). To this end, we shift one series by $\Delta t$ and find the respective Pearson correlation coefficient. The data show a correlation maximum with Esrange and Pallas at $+3\,\mathrm{h}$ and $+1\,\mathrm{h}$ with Jergul/Karasjok. This means that these lag behind Svanvik. Of all stations, only Pallas displays a sufficiently high correlation with Svanvik ($r^2 \geq 0.61$) (Mills et al., 2020, Section 12.5). We, therefore, choose Pallas as the reference station for the following reconstruction procedure. We derive a projection of the generalized ozone climatology to Svanvik and account for the time lag by shifting the climatology by $3\,\mathrm{h}$:

$$P_{\mathrm{Svanvik}} = \frac{\langle[O_3]\rangle_{\mathrm{hourly}}^{\mathrm{Svanvik}}}{\langle[O_3]\rangle_{\mathrm{hourly},\,t'=t-3}}. \tag{6}$$

We apply Eq. (5) to derive 1-hourly anomalies compared to the generalized climatology for each active station in 2018

$$\Delta[O_3]_{\mathrm{hourly}}{}^{i} = [O_3]_{\mathrm{hourly}}{}^{i} - \langle[O_3]\rangle_{\mathrm{hourly}}, \tag{7}$$

with $i \in \{\text{Esrange, Pallas, Svanvik}\}$.

Observational data for Svanvik, Esrange, and Pallas for July 2018 is depicted in Fig. 6a. For reference, we overlay the generalized climatology, the climatology for Svanvik in 1-hourly resolution, and indicate the time-lag corrected generalized climatology.

We also correct the derived ozone anomalies at Pallas for the time lag $\Delta[\mathrm{O_3}]_{\mathrm{hourly,\,t-3}}^{\mathrm{Pallas}}$ and use the projection (Eq. (6)) to reconstruct anomalies for the missing values at Svanvik:

$$\Delta[\mathrm{O_3}]_{\mathrm{hourly}}^{\mathrm{Svanvik,\,reco}} = \Delta[\mathrm{O_3}]_{\mathrm{hourly,\,t-3}}^{\mathrm{Pallas}} \cdot P_{\mathrm{Svanvik}}. \tag{8}$$

The result is depicted in Fig. 6b, where the 1-hourly ozone concentration anomalies are shown together with the reconstructed anomalies for Svanvik. We do not account for the transport of pollutants or advection of ozone in our reconstruction procedure which results in a prominent lag between the reconstruction and the observations on July 25/26. In the context of risk assessment of ozone damage on vegetation, this has no large impact, as the applied flux-based metric $\mathrm{POD_y}$ is usually integrated over a whole season (e.g. Mills et al., 2011).

Finally, we add these anomalies to the Svanvik climatology, account for the estimated bias due to the change in ground-level background ozone ($\delta[\mathrm{O_3}] = 1.2\,\mathrm{ppb}$), and derive the reconstructed time series

$$[\mathrm{O_3}]_{\mathrm{hourly}}^{\mathrm{Svanvik,\,reco}} = \langle[\mathrm{O_3}]\rangle_{\mathrm{hourly}}^{\mathrm{Svanvik}} + \Delta[\mathrm{O_3}]_{\mathrm{hourly}}^{\mathrm{Svanvik,\,reco}} + \delta[\mathrm{O_3}]. \tag{9}$$

In Fig. 6c, our reconstruction is shown together with the observed data before and after the gap and the CAMSRAQ evaluated at the nearest neighboring grid point. Both perform qualitatively well. To quantify the performance of our reconstruction and the CAMSRAQ, we compute RMSEs for the days in July for which observational data is available. We find a $\mathrm{RSME} = 8.20\,\mathrm{ppb}$ for our reconstruction and $\mathrm{RSME} = 7.52\,\mathrm{ppb}$ for the CAMSRAQ. This indicates that our reconstruction has an accuracy of about $76\%$ and its performance is comparable with CAMSRAQ ($80\%$) despite not accounting for atmospheric transport and chemical transformation explicitly. For comparison, the computed accuracy of data taken at Pallas in 2018 without further processing is decent ($69\%$) while data taken at Svanvik in July 2019 agrees fairly well ($72\%$).

## 4   Discussion & conclusions

We derived a representative ozone climatology for northern Fennoscandia based on long-term ground-level ozone monitoring in northern Finland, Norway, and Sweden. Based on this generalized ozone climatology, we assessed the quality of available global (MACC and CAMSRA) and regional (CAMSRAQ) reanalysis products for northern Fennoscandia focussing on the seasonality of ozone. We confirm previously published results concerning the quality of global reanalysis products (Huijnen et al., 2020; Barten et al., 2020) and find that the observed ozone patterns in northern Fennoscandia are not reproduced well. Better performance was displayed by the regional model reanalysis CAMSRAQ ensemble which reproduces the observed ozone seasonality well, although with a remaining annual average deviation of up to $-7\,\mathrm{ppb}$. All products showed deficits, in particular during winter and spring. Spatial patterns of deviation from the generalized climatology indicate a substantial underestimation of ozone abundance in the global reanalysis products on the west coast of northern Fennoscandia. This could be due to their spatial resolution, e.g. a high oceanic fraction in the coastal grid cells or representation of elevation. We confirm that a higher spatiotemporal resolution, assimilation of vertical ozone profiles, and if applicable assimilation of in situ observations at ground-level lead to better constrained reanalysis products, especially at high latitudes during times when the coverage by passive sounders onboard satellites is low.

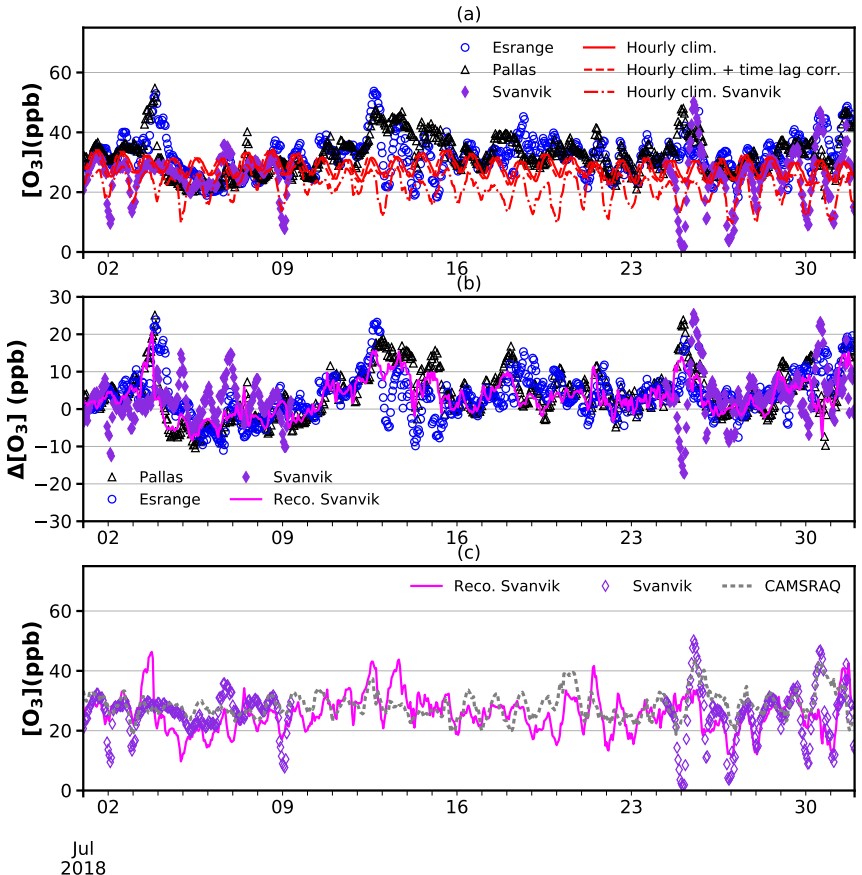

**Figure 6.** Reconstruction procedure for missing [O₃] data (July 9–23, 2018). Observed 1-hourly [O₃] are shown together with 1-hourly climatologies derived for northern Fennoscandia (combined data from Esrange, Jergul/Karasjok, Pallas) and Svanvik. (a) Time series supplemented with 1-hourly climatologies. The time-lag correction of the northern Fennoscandia climatology is also indicated; (b) observed and reconstructed anomalies; (c) reconstructed [O₃] for Svanvik in comparison with CAMSRAQ evaluated at the nearest neighboring grid point.

There is a multitude of probable reasons for the differences found between the reanalysis products and observations. The enhancements which led from the MACC reanalysis to CAMSRA have been reported and discussed by Inness et al. (2019) on global scales. Amongst others, assimilation of ozone profiles from satellite retrieval (compared to column densities) and an upgraded ozone chemistry have led to an enhanced performance of CAMSRA, but a considerable bias remains (Huijnen et al., 2020). In particular, Barten et al. (2020) reported a pronounced underestimation for CAMSRA in the high Arctic (e.g., Summit, Greenland) and attribute this to an insufficient representation of a mechanistic dry deposition scheme to the ocean. The large deviation which we found in all seasons but summer either points to a deficit in modeled removal processes or too weak model constraints by data from passive sounders onboard satellites in polar winter. In particular, too high dry deposition velocities over snow and ice-covered surfaces would not allow for a sufficient build-up of ozone and precursors in winter

leading to too low modeled ozone concentrations (Falk and Sinnhuber, 2018; Falk and Søvde Haslerud, 2019). Due to the higher spatial resolution of the regional air quality models, CAMSRAQ is capable of capturing small-scale depletion and peak episodes of ozone. The higher spatial and temporal resolution improves daily and seasonal cycles of modeled ozone which is especially important for the use in risk assessment for vegetation damage and human health. Improvements in atmospheric transport as part of the OpenIFS updates may play also a role but cannot be assessed from our analysis. The higher spatial and

temporal resolution of CAMSRAQ aside, we can assume the assimilated ground-level ozone data was another driver for the different performance as passive sounders onboard satellites typically resolve $[O_3]$ at the surface rather poorly and hence do not constrain the global models well enough (Andersson et al., 2017).

To account for missing data from the 2018 record at Svanvik located in northern Norway, we proposed a routine for reconstruction of 1-hourly ozone data adhering to the UNECE-LRTAP conventions (Mills et al., 2020). We performed a Reynolds

decomposition into anomalies and climatology, identified a reference station with the highest Pearson correlation coefficient, synchronized the time series using a time lag correlation, and corrected for a bias induced by the increase in ground-level background ozone concentrations since the end of the regular measurements at Svanvik in the mid-1990s. As we don't take atmospheric transport of pollutants into account, the reconstructed data display inaccuracies in the timing of peak episodes. This deficit, however, has no large impact in the context of risk assessment of ozone damage on vegetation because the applied

flux-based metrics typically integrate the ozone uptake over a whole season. Our devised reconstruction method's performance ($76\%$ accuracy) is compatible with evaluating CAMSRAQ at the nearest neighboring grid point ($80\%$) and better than standard methods ($69-72\%$). However, two criteria have to be met before our reconstruction can be performed: 1. Availability of overlapping long-term series. 2. Availability of overlapping data from a reasonably close-by site with a high Pearson correlation coefficient during the occurrence of the gap.

We have shown that the representation of ground-level ozone concentration in the global state-of-the-art reanalysis product CAMSRA is poor in winter but good in summer. In all seasons but summer, negative deviations occur over northern Fennoscandia. In summer, CAMSRA displays a pronounced bias towards higher than observed ozone concentrations ($6\,\mathrm{ppb}$) in regions east of the Scandinavian Mountains. The regional reanalysis product CAMSRAQ displays slightly too low ozone concentrations throughout all seasons, though, not significant in summer. To assess the impact of ozone on vegetation risks, we computed

a relative Cumulative Uptake of Ozone. Positive deviations in $[O_3]$ in summer compared to the generalized climatology for northern Fennoscandia cause a relative percentage deviation of CAMSRA of $17\%$. For the MACC reanalysis, we find $7\%$. The lower deviation does not indicate better performance but is due to the pronounced underestimation of $[O_3]$ in spring countering too high ozone abundances in summer. This is also reflected by diverging results for coniferous and deciduous trees. CAMSRAQ deviates by only $2\%$ confirming its suitability for vegetation risk assessments. Our results are in line with Hayes

et al. (2018) who showed that a climate change-induced increase in summer ground-level ozone concentration can affect the stomatal uptake of ozone in southwestern Sweden in the order of $3-16\%$. While environmental conditions in spring and fall limit the effects for most species except for coniferous species which are photosynthetically active at low temperatures and could be moderately affected.

Our devised gap-filling method is to be preferred over data from close-by stations or data from the same period but different
370   years. Overall, CAMSRAQ showed the best performance. We can therefore recommend using the CAMSRAQ for gap-filling
of ozone monitoring data. It is also a valid choice for ozone risk assessment on vegetation in northern Fennoscandia. Global
reanalysis products are not recommended for this purpose.

*Code availability.* Python 2.7 code is available under Common Creative Licence on the GitHub repository of the corresponding author.
Please get in touch for further information.

*Data availability.* All observational data is available form NILU's EBAS database. MACC and CAMS reanalysis data is available through
ECMWF's data services. CAMSRAQ is available from the regional atmosphere data service of copernicus.

*Author contributions.* SF aquired and processed all data, conducted the analysis, and composed the figures and manuscript. AVV and FS
contributed to proofreading. All authors contributed to discussion.

*Competing interests.* The authors declare that they have no conflict of interest.

*Acknowledgements.* This work was funded by the Norwegian Research Council (NRC) through the project The double punch: Ozone and
climate stresses on vegetation (268073). We would also like to thank the LATICE research group and the EMERALD project (294948) for
supporting this work.

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
