# Peer review of "Technical note: Quality assessment of ozone reanalysis products and gap-filling over subarctic Europe for vegetation risk mapping"

_Atmospheric Chemistry and Physics, 2021_

## Author Comment (AC2)

Atmos. Chem. Phys. Discuss., referee comment RC2
https://doi.org/10.5194/acp-2021-527-RC2, 2021

[Figure]

**Comment on acp-2021-527**

Anonymous Referee #2
* * *
Referee comment on "Technical note: Quality assessment of ozone reanalysis products over subarctic Europe for biome modeling and ozone risk mapping" by Stefanie Falk et al., Atmos. Chem. Phys. Discuss., https://doi.org/10.5194/acp-2021-527-RC2, 2021
* * *
**general comments**

This study looks at a way to fill in gaps in ground ozone data, principally in the subarctic zone. It uses the example of several data stations, some that stopped measuring in the 1990s, one that was resurrected for 2018/2019 and 2 that are still measuring at present. It presents 3 different reanalysis products that could be used for gap filling, finally settling for CAMSRAQ. It shows the statistical analysis for assessing closeness to the seasonal cycle and being able to fill the gaps in measurement data.

I definitely recommend it for publication with only minor changes.

**specific comments**

This paper is easy to follow and well written with a very thorough introduction and a good justification for why this analysis is appropriate to Northern Fennoscandia. It does not go as far as recommending how it could be used in other global regions, but justifies that this analysis is particularly applicable to these remote areas with large seasonal variations.

For those unfamiliar with reanalysis products, we assume that these 3 products are the only ones available, 2 global and 1 regional. Maybe the range of other products could be introduced and justify why these 3 were chosen.

The word Biome in the title is never used again, maybe just ozone risk products is needed? Or vegetation ozone risk products?

There seemed to be no explanation of how missing data during a forest fire period may be harder to fill than during a more normal period. Maybe this technique needs to be applied when something serious like a forest fire impedes entry to the station and data is lost.

The analysis of the seasonality is interesting in itself- showing how it is still hard to model and predict.

Figure 6 is the final and most important figure. It should be put before the conclusions, otherwise it may be missed! On line 278, you state that your devised method performs better (78% accuracy) than CAMSRAQ at nearest neighbour. This is very important and is

stated in the abstract too but you could compare it to the other methods too. How much better is it?

**technical corrections**

Line 19 - O3 "acts" as a potent greenhouse gas

Line 92 "data taking" and line 272 "data taking" should be replaced by "Measurements"

Figure 3- The generalized ozone climatology shown as "a" gray band represents------ On average, all reanalysis products "underestimate" [O3].

Line 172- Tromsø "where" [O3]

Line 188 - larger negative deviation from "observations" in DJF and MAM. This indicates that CAMSRAQ might have different issues depending on the region of interest. "Different issues or different uncertainties- maybe this could be elaborated?"

Conclusions Line 248

You say "We confirm that a high spatial and temporal resolution, state-of-the-art mechanistic removal processes (land–atmosphere–ocean), and assimilation of in situ observations at ground-level are a must to constrain reanalysis products," but have you really confirmed it or explained that the land–atmosphere–ocean interaction is applied here?

Line 263 – "updates may "also" play a role"

---

## Author Comment (AC3)

**Authors' response**

To acp-2021-527-RC2 (23 Aug 2021): We thank the anonymous referee #2 for their comments. We will address the specific comments in detail in our final revision of the manuscript. Here, we shall give a brief response to questions where it seems appropriate. We have already tended to the minor, technical comments.

**Specific comments**

- This paper is easy to follow and well written with a very thorough introduction and a good justification for why this analysis is appropriate to Northern Fennoscandia. It does not go as far as recommending how it could be used in other global regions, but justifies that this analysis is particularly applicable to these remote areas with large seasonal variations. Thank you.

- For those unfamiliar with reanalysis products, we assume that these 3 products are the only ones available, 2 global and 1 regional. Maybe the range of other products could be introduced and justify why these 3 were chosen. Thank you for pointing this out. Following the advises of referee #1, we add a paragraph in the final revision regarding other available reanalysis products and why these three (MACC, CAMSRA, CAMSRAQ) were chosen.

- The word Biome in the title is never used again, maybe just ozone risk products is needed? Or vegetation ozone risk products? We elaborate on the title and change it to *Technical note: Quality assessment of ozone reanalysis products and gap-filling over subarctic Europe for vegetation risk mapping*

- There seemed to be no explanation of how missing data during a forest fire period may be harder to fill than during a more normal period. Maybe this technique needs to be applied when something serious like a forest fire impedes entry to the station and data is lost. Thank you for this remark.
  As we described, gap filling is usually done by using mean values from the same period from previous years or by using mean values from the same time of day on previous days. If forest fires are rare, those mean values will not be good candidates for gap filling. Data from a reference station selected based on a high correlation factor alone is not sufficient, because a correlation does not account for systematic offsets or the transport of pollutants.

  We add a paragraph in the appropriate section.

- The analysis of the seasonality is interesting in itself- showing how it is still hard to model and predict. Indeed, this was the most astonishing result of our analysis which has implications even beyond vegetation risk mapping. The ozone concentration affects the oxidation capacity and halogen chemistry of the atmosphere. Hence, e.g., the rate at which mercury is processed and deposited in the Arctic (see Section 2.5.1.3. in the AMAP Assessment 2011: Mercury in the Arctic).

- Figure 6 is the final and most important figure. It should be put before the conclusions, otherwise it may be missed! Thank you this is probably subject to the final typesetting and cannot be taken care of at this stage.

- On line 278, you state that your devised method performs better (78% accuracy) than CAMSRAQ at nearest neighbour. This is very important and is stated in the abstract too but you could compare it to the other methods too. How much better is it? Given that we already confirmed a high correlation with observations at Pallas. The only other methods in accordance to ICP Vegetation, would be to compare with the data from Pallas directly without further processing (accuracy 69 %) or comparing with, e.g., the July 2019 at Svanvik (72 %). We will include these in the final revision of the manuscript.
  Thanks to your comment, we went through our calculation of mean accuracy again and found a shortcoming. We determine, the accuracy of our method to 76 % and for CAMSRAQ to 80 %. We account for this in the final revision and rephrase our abstract and conclusion accordingly. This result is actually more consistent with our expectations. As we have not taken chemical transformation and transport into consideration in our method.
  This mistake was unintentional and does not affect the overall content of the manuscript.

**Minor comments**

- Line 19 - O3 "acts" as a potent greenhouse gas Done.

- Line 92 "data taking" and line 272 "data taking" should be replaced by "Measurements" Done.

- Figure 3- The generalized ozone climatology shown as "a" gray band represents—— On average, all reanalysis products "underestimate" [O3]. Done.

- Line 172- Tromsø "where" [O3] Done.

- Line 188 - larger negative deviation from "observations" in DJF and MAM. Done.

- This indicates that CAMSRAQ might have different issues depending on the region of interest. "Different issues or different uncertainties- maybe this could be elaborated?" Thank you for pointing this out. We elaborate on the sentence and rephrase: *This indicates that underlying uncertainties in CAMSRAQ manifest differently at higher latitudes. Enhancements that lead to better model performances in mid latitudes, hence, do not necessarily affect results in the Arctic and subarctic in the same way.*

- Conclusions Line 248: You say "We confirm that a high spatial and temporal resolution, state-of-the-art mechanistic removal processes (land–atmosphere–ocean), and assimilation of in situ observations at ground-level are a must to constrain

reanalysis products," but have you really confirmed it or explained that the land–atmosphere–ocean interaction is applied here? You are right. We have, in fact, not proven that in particular. What we confirm is that *the assimilation of vertically ozone profiles, if applicable ground-level observations, and a higher spatiotemporal resolution lead to better constrained reanalysis products.*

- Line 263 – "updates may "also" play a role" Done.

---

## Author Response (AR1)

**Final response**

We thank all anonymous referees for their constructive and positive evaluation of our manuscript.

We have addressed all minor comments already during the discussion phase. For reference, we include our responses at the end of this document. The related changes clearly emerge from the track-changes file. Regarding the specific comments, we refer to our previous answers (text typeset in *italic*) and give additional details about the changes that we incorporated in the manuscript.

**To Ref #1**

- The title suggests a specific focus on biome modeling and ozone risk mapping which is also properly introduced in the Introduction section. However, throughout the rest of the manuscript this focus is lost and is shifted towards a quantitative assessment of the reanalysis products and the gap-filling method with minor discussion points on the further implication on biome modeling and ozone risk mapping. I suggest the authors to revise the title or to, preferably, put additional effort in quantifying the effects of the use of these different reanalysis products (and also gap-filling techniques) on biome modeling. For example: This manuscript shows that the representation of CAMSRA is poor in winter, but good in summer which coincides with the growing season and maximum in ozone uptake (e.g. Hayes et al. (2019)). The resulting effect on integrated flux quantities such as $POD_y$ might therefore be limited. *We will address the implications on integrated ozone flux quantities more thoroughly in our discussions and conclusions in the final revision of the manuscript. We consider consulting Fig. 4 to consolidate implications on the biome (spatial) level.*
  Following the advice of Ref #1, we estimate the effect on an integrated ozone flux quantity and include this in an additional Section (3.2. Implications on integrated flux quantities) with the following content. We also add a remark in our introduction:
  "We look at the respective seasonal cycles, spatial patterns, and derive the relative impact on an integrated-flux metric."

As the computation of POD is non trivial, we make some simplifications. We choose Svanvik as example location and compute a Cumulative Uptake of Ozone (CUO) with a threshold of $y = 0$ that is defined as follows

$$\text{CUO}_0 = \sum \Phi^{O_3}(t) \cdot \Delta t. \tag{1}$$

The ozone flux through the stomata is given by

$$\Phi^{O_3}(t) = [O_3](t) \cdot g_{\text{sto}}(t) \cdot \frac{r_c}{r_c + r_b}. \tag{2}$$

Ozone concentrations $[O_3]$ have to be converted from ppb to $\text{mmol}\,\text{m}^{-3}$ using the ideal gas law ($V^{-1} = \frac{P}{RT}$) and multiplied with $10^{-6}$. Multiplying with $60^2$

converts the flux to $\mathrm{mmol\,m^{-2}\,h^{-1}}$. For simplicity, we assume standard pressure ($P_{\mathrm{std}} = 1.013 \cdot 10^{-5}\mathrm{Pa}$) but insert observed 2018 temperatures at Svanvik. We neglect the quasi laminar ($r_b$) and leaf surface resistance ($r_c$) terms. This can be justified by only looking at the percentage differences in the following and not the absolute CUO values. The stomatal conductance $g_{\mathrm{sto}}$ follows from a Jarvis model like approach

$$g_{\mathrm{sto}} = g_{\mathrm{max}} \cdot f_{\mathrm{light}} \cdot \max\left\{f_{\mathrm{min}}, f_T \cdot f_{\mathrm{VPD}}\right\}. \tag{3}$$

For the parameters with respect to light ($f_{\mathrm{light}}$), temperature ($f_T$), vapor pressure deficit ($f_{\mathrm{VPD}}$), and minimum conductance ($f_{\mathrm{min}}$), we refer to the LRTAP conventions on critical levels for vegetation.

We use meteorological data (temperature, humidity, global irradiance) from 2018 observed at Svanvik and compute CUO for parameterizations of boreal deciduous and coniferous trees (Table III.11, in LRTAP conventions). We compare CUO computed with the climatologies based on MACC, CAMSRA, and CAMSRAQ with CUO based on the bias corrected ozone climatology for Svanvik and find

Table 1: Relative difference in percent between reanalysis products and observed ozone by means of CUO0 computed for boreal parameterizations of deciduous and coniferous trees.

| Reanalysis product | deciduous | coniferous |
|---|---|---|
| MACC | 8.1 | 6.4 |
| CAMSRA | 17.15 | 17.12 |
| CAMSRAQ | 2.0 | 1.9 |

All reanalysis products overestimate CUO compared to observations. CAMSRAQ shows the best performance with a $2\,\%$ deviation. Whereas CAMSRA overall represents the seasonal cycle better, its performance with respect to CUO ($17\,\%$) is worse than MACC ($7\,\%$). This can be attributed to the pronounced bias towards higher ozone concentrations in CAMSRA during summer as emerges from Fig. 4.

Based on these results, we add a paragraph to our discussion:

"We have shown that the representation of ground-level ozone concentration in the global state-of-the-art reanalysis product CAMSRA is poor in winter but good in summer. In all seasons but summer, negative deviations occur over northern Fennoscandia. In summer, CAMSRA displays a pronounced bias towards higher than observed ozone concentrations ($6\,\mathrm{ppb}$) in regions east of the Scandinavian Mountains. The regional reanalysis product CAMSRAQ displays slightly too low ozone concentrations throughout all seasons, though, not significant in summer. To assess the impact of ozone on vegetation risks, we computed a relative Cumulative Uptake of Ozone. Positive deviations in $[\mathrm{O_3}]$ in summer compared to

the generalized climatology for northern Fennoscandia cause a relative percentage deviation of CAMSRA of 17%. For the MACC reanalysis, we find 7 %. The lower deviation does not indicate better performance but is due to the pronounced underestimation of $[O_3]$ in spring countering too high ozone abundances in summer. This is also reflected by diverging results for coniferous and deciduous trees. CAMSRAQ deviates by only 2% confirming its suitability for vegetation risk assessments. Our results are in line with Hayes eta al. (2019) who showed that a climate change-induced increase in summer ground-level ozone concentration can affect the stomatal uptake of ozone in southwestern Sweden in the order of $3-16\%$. While environmental conditions in spring and fall limit the effects for most species except for coniferous species which are photosynthetically active at low temperatures and could be moderately affected."

And refine:
"[CAMSRAQ] is also a valid choice for ozone risk assessment on vegetation in northern Fennoscandia. Global reanalysis products are not recommended for this purposes."

- It is unclear why specifically these 3 ozone reanalysis products have been chosen to include in this study and other tropospheric ozone reanalysis products such as TCR-2 or JRA-55 have been excluded. See for example Huijnen et al. (2020) and Park et al. (2020) for global and regional application of these reanalysis products including comparisons with CAMSRA respectively. Especially the use of the MACC reanalysis data is questionable. This product has already been identified as less accurate compared to CAMSRA in other studies (e.g. Inness et al. (2019)) and is, as far as I am aware, not supported anymore because it is replaced by the CAMSRA system. *The inclusion of the MACC reanalysis has historical reasons. The MACC reanalysis is still well known in the wider community. Although its lower accuracy compared to CAMSRA has been recently shown. To assess whether and how the improvements to the CAMS assimilation system affect the reanalysis results in our focus area, we kept the analysis of the MACC reanalysis after switching to the newer CAMSRA as it became available. CAMSRAQ has been specifically chosen to test whether a higher spatiotemporal resolution will also show better results in our focus area. The more general inter-comparison studies (eg. Huijnen et al. (2020)) did not look into the seasonal cycle in detail but compared seasonal averages. These seasonal averages, however, suggest a similar performance of CAMSRA and TRC-2 in our focus area, therefore we assume our selection to be representative for the state-of-the-art global reanalysis products. Thank you for pointing out the comprehensive JRA-55 reanalysis which is very interesting in terms of climatological studies due to it's length. With a horizontal resolution of T319 it may not perform substantially better than CAMSRA. As we have shown, only the regional reanalysis ensemble (CAMSRAQ) performs reasonably well through all seasons. Furthermore, according to the JRA-55 handbook (Section 4.1.10 and 5.1) the atmospheric mixing ratios of ozone are only available in 6-hourly temporal resolution and interpolated*

*to pressure levels (e.g. 1000 hPa). Both disqualify the JRA-55 as substitution for observational data in our case, because the computation of* $POD_y$ *requires 1-hourly* $[O_3]$ *input.*

Accordingly, we added the following to Section 2.2:

"The MACC reanalysis is still well known and used in the wider community, albeit its lower accuracy compared to CAMSRA (Huijnen et al. 2020). To assess whether and how the improvements to the CAMS assimilation system affect the reanalysis results in our focus area, we analyze both MACC and CAMSRA. CAMSRAQ has been specifically chosen to test whether a higher spatiotemporal resolution will also give better results in our focus area.

On global scales, at least two other ozone reanalysis products are available, the Tropospheric Chemistry Reanalysis (TRC) 1 and 2 (Miyazaki et al. 2020) and the Japanese Reanalysis 55 (JRA-55) (Kobayashi et al. 2015). As part of the comprehensive reanalysis inter-comparison study by Huijnen et al. (2020), TRC-1 and 2, CAMS interim reanalysis, and CAMSRA were compared by means of seasonal averages. The results suggested a similar performance of CAMSRA and TRC-2 in our focus area. Therefore, we assume our selection to be representative for the state-of-the-art global reanalysis products. Therefore, we assume our selection to be representative for the state-of-the-art global reanalysis products.

The comprehensive JRA-55 reanalysis is the longest reanalysis dataset available spanning several decades. With a horizontal resolution of T319, 6-hourly temporal resolution and interpolated to pressure levels (e.g. 1000 hPa) it is too coarse and not suitable for our purpose."

- For CAMSRAQ, the period of 2014-2018 is used to compute the daily mean ozone climatologies compared to the period 2003-2012 for the MACC and CAMSRA products. The authors should discuss if and how this relatively short period affects the computed climatologies (also with respect to observations which cover an even longer period) also considering the anomalous summer of 2018 as the authors show in their Fig. 6b. *We will elaborate on this in the final revision.*

    *The length of the CAMSRAQ is indeed too short to compute a reliable climatology in a purely statistical sense, though interannual variability in the reanalysis may be considered to be somewhat lower than in actual ground-level observations (e.g., systematic errors in instruments). In statistical terms, a systematical underestimation of the CAMSRAQ cannot be demonstrated, only suggested. Assuming that background ozone concentrations are indeed increasing, the reanalysis based climatologies are biased towards higher annual average ozone concentrations at ground-level than the observational climatologies. This does not affect the main issue of the global reanalyses not reproducing the seasonal cycle.*

    We add the following to Section 3:

"However, the reanalysis products' time series are not sufficiently long enough to study deviations from the observed climatology with a high statistical significance. Though, the associated standard deviation is usually smaller in models compared to observations due to the inherent spatiotemporal averaging. This has no impact on our qualitative results. Recent analyses indicate a levelling or decline of tropospheric background $[O_3]$ over Europe after 2007 (Cooper et al. 2014; Wespes et al. 2018;Gaudel et al. 2018) following a steady increase over the past decades (IPCC - Intergovernmental Panel on Climate Change, 2013, Chapter 2). This indicates that the observation-based generalized northern Fennoscandia climatology which includes data before 2007 could be biased towards higher annual average $[O_3]$. As estimated in Section 2.1, the climatology derived for Svanvik is insignificantly underestimating present day $[O_3]$"

- In the results Section the authors show the divergence of reanalysis products from generalized ozone climatology for northern Fennoscandia spatially (Fig. 4 and accompanied text in line 167-179). The main analysis of this subsection mainly focuses on the seasonal skill scores (from RMSE) which can also be derived from Fig. 3, rather than the spatial patterns in the divergence. Furthermore, the need for presenting these spatial patterns appears to be limited also because they do not play a prominent (if any) role in the Abstract and Conclusions. The authors should more strongly motivate and discuss these spatial patterns or remove Fig. 4 and combine the text with the analysis presented in line 147-166. *We may consider removing Fig. 4 and combine the text as suggested or use it for elaborate on the implication of our findings on biome (spatial) level.*

We elaborate on the description of Fig. 4:

"In Fig.4, the seasonally averaged divergence is shown between each reanalysis product and the generalized ozone climatology which shall represent the expected ground-level ozone background for the whole region. We also computed the root mean square error (RMSE) over land-only which is displayed in the upper left corner of the respective panel. As expected, the global reanalysis products, MACC and CAMSRA (Fig. 4a,b), show substantial negative deviations ($\Delta[O_3] < -10\,\mathrm{ppb}$) in winter (DJF) and spring (MAM). The respective RSMEs range between $(12.3 - 15.2)\,\mathrm{ppb}$ (MACC) and $(10.1 - 15.6)\,\mathrm{ppb}$ (CAMSRA). The smallest deviations ($\Delta[O_3] > -4\,\mathrm{ppb}$) occur in summer (JJA). In Summer, the MACC reanalysis deviations are overall negative except for a small region east of Tromsø where $\Delta[O_3]$ are slightly positive (RSME = $3.9\,\mathrm{ppb}$). While a positive $[O_3]$ deviation would be expected over the Scandinavian Mountains due to the higher elevation compared to the reference height of the generalized Fennoscandia climatology, the spatial pattern of the MACC reanalysis displays lower $[O_3]$ in coastal areas in the west which could point to an influence of oceanic fractions in these grid cells. The lowest deviation occurs in areas with a mean elevation similar to the generalized climatology for northern Fennoscandia. Especially in Summer, CAM-

SRA shows a distinctive gradient with positive divergence furthest East, in areas surrounding the northern Gulf of Boothia. Similar to MACC, coastal areas in the west seem to be influenced by oceanic fractions in these grid cells. The divergence of CAMSRAQ from the generalized ozone climatology is considerably smaller than for the global reanalysis products and stays below $20\,\%$ (RSME $\leq 6.6\,$ppb) at all times. The white areas at the northern and eastern borders represent the domain borders (Fig. 4c). The largest deviations are again found in winter and spring, while the smallest occur in summer (RSME $\leq 2.6\,$ppb)). The divergence in ozone follows the terrain more closely. Consistent with the on average too low ozone abundance, the highest negative deviations are displayed in areas that lie at a lower elevation than the reference stations of the generalized climatology."

We add to the abstract:
"Spatial patterns suggest a substantial underestimation of ozone abundance in the global reanalysis products on the west coast of northern Fennoscandia probably due to spatial resolution."

And include in the conclusions:
"This is could be due to their spatial resolution, e.g. a high oceanic fraction in the coastal grid cells or representation of elevation."

**To Ref #2**

- For those unfamiliar with reanalysis products, we assume that these 3 products are the only ones available, 2 global and 1 regional. Maybe the range of other products could be introduced and justify why these 3 were chosen. *Thank you for pointing this out. Following the advises of referee #1, we add a paragraph in the final revision regarding other available reanalysis products and why these three (MACC, CAMSRA, CAMSRAQ) were chosen.* See answer to Ref #1.

- The word Biome in the title is never used again, maybe just ozone risk products is needed? Or vegetation ozone risk products? We elaborate on the title and change it to *Technical note: Quality assessment of ozone reanalysis products and gap-filling over subarctic Europe for vegetation risk mapping*

- There seemed to be no explanation of how missing data during a forest fire period may be harder to fill than during a more normal period. Maybe this technique needs to be applied when something serious like a forest fire impedes entry to the station and data is lost. Thank you for this remark.
  *As we described, gap filling is usually done by using mean values from the same period from previous years or by using mean values from the same time of day on previous days. If forest fires are rare, those mean values will not be good candidates for gap filling. Data from a reference station selected based on a high correlation*

*factor alone is not sufficient, because a correlation does not account for systematic offsets or the transport of pollutants.*

We add the following paragraph to Section 3.3 (formerly 3.2):
"As described in Section 1, gap-filling is usually done by using mean values from the same period from previous years or by using mean values from the same time of day from previous days. Considering forest fires are rare events, those mean values will not be good candidates for gap-filling. In addition, data from a reference station selected based on a high correlation factor alone is not sufficient, because a correlation does not account for systematic offsets or the transport of pollutants."

- The analysis of the seasonality is interesting in itself- showing how it is still hard to model and predict. *Indeed, this was the most astonishing result of our analysis which has implications even beyond vegetation risk mapping. The ozone concentration affects the oxidation capacity and halogen chemistry of the atmosphere. Hence, e.g., the rate at which mercury is processed and deposited in the Arctic (see Section 2.5.1.3. in the AMAP Assessment 2011: Mercury in the Arctic).*

- Figure 6 is the final and most important figure. It should be put before the conclusions, otherwise it may be missed! *Thank you this is probably subject to the final typesetting and cannot be taken care of at this stage.*

- On line 278, you state that your devised method performs better (78% accuracy) than CAMSRAQ at nearest neighbour. This is very important and is stated in the abstract too but you could compare it to the other methods too. How much better is it? *Given that we already confirmed a high correlation with observations at Pallas. The only other methods in accordance to ICP Vegetation, would be to compare with the data from Pallas directly without further processing (accuracy 69 %) or comparing with, e.g., the July 2019 at Svanvik (72 %). We will include these in the final revision of the manuscript.*
  *Thanks to your comment, we went through our calculation of mean accuracy again and found a shortcoming. We determine, the accuracy of our method to 76 % and for CAMSRAQ to 80 %. We account for this in the final revision and rephrase our abstract and conclusion accordingly. This result is actually more consistent with our expectations. As we have not taken chemical transformation and transport into consideration in our method.*
  *This mistake was unintentional and does not affect the overall content of the manuscript.*

  We rephrase the paragraph in Section 3.3 accordingly:
  "We find a RSME = 8.20 ppb for our reconstruction and RSME = 7.52 ppb for the CAMSRAQ. This indicates that our reconstruction has an accuracy of about 76 % and its performance is comparable with CAMSRAQ (80 %) despite not accounting for atmospheric transport and chemical transformation explicitly. For comparison, the computed accuracy of data taken at Pallas in 2018 without further processing is decent (69 %) while data taken at Svanvik in July 2019 agrees fairly well (72 %)."

We add to our conclusions and rephrase:

"Overall, CAMSRAQ showed the best performance. We can therefore recommend using the CAMSRAQ as the first choice for gap-filling of ozone monitoring data. It is also a valid choice for ozone risk assessment on vegetation in northern Fennoscandia. Our devised method is to be preferred over data from close-by stations or data from the same period but different years. Global reanalysis products are not recommended for this purpose. "

**To minor comments Ref #1 and #2**

- page 1, line 1: "regional or global" → "regional and global". We follow the suggestion.

- page 3, line 74: "such as a" → "such as". We follow the suggestion.

- page 6, Fig. 2 label: "6.6. ppb" → "6.6 ppb". We corrected the typo.

- page 6, Fig. 2 label: "The magnitude ... late summer". In my opinion this does not belong in a label also because it is repeated in the main text. We follow the advice and remove the additional information in the caption.

- page 6, line 140: "In the following". Add subject. ?

- page 7, line 146: "ozone reanalysis product" → "ozone reanalysis products" We fix this typo.

- page 7, line 161: "This indicates an insufficient vertical resolution of these models". What is the vertical resolution of these datasets (e.g. the height of the surface layer)? To be included in the methods (Table 2). We update Table 2 accordingly. MACC and CAMSRA vertical levels amount to 60 levels and are the same as for Integrated Forecast System (IFS); the level thickness at surface is 10 m. For CAMSRAQ, the vertical levels vary for each ensemble member, data is given at the actual surface level.

- page 7, line 166: "observation" → "observations" or "observed". We correct the spelling to *observations*.

- page 8, Fig. 3: I suggest using same xticks as in Fig. 2 for better comparison between the two. We adjust the xticks as suggested and update the figure in the next revision of our manuscript.

- page 8, Fig. 3 label: "The global ... low [O 3 ]." . In my opinion this does not belong in a label also because it is repeated in the main text. We folllow the advice and remove the text from the caption.

- page 8, line 167: The term "tropospheric ozone background" has been used throughout the manuscript (at multiple instances before and after this line). As this study deals with the ground-level ozone climatology I ask the authors to consider the

terminology "ground-level ozone background" to avoid ambiguity. Thank you for pointing this out. We have changed the term as suggested through out the manuscript.

- page 9, Fig. 4: The colorbar-label is cut off. For (a) and (b) also the xlabel "Longitude" is cut off just short. For all other Figures the sublabels are located above the panel while forthis Figure they are located below the panel. Furthermore, the labels could benefit from some extra dpi if possible. As suggested by the referee, we consider removing this figure from the manuscript.

- page 10, line 183-185: "given in ... of ppb". Can be removed. We remove these sentences, as suggested.

- page 12, line 233: Remove closing bracket. Done.

- page 12, line 236: "an RMSE" → "a RMSE". Done.

- page 12, line 239: "Conclusions" → "Discussion & Conclusions". Done.

- Line 19 - O3 "acts" as a potent greenhouse gas Done.

- Line 92 "data taking" and line 272 "data taking" should be replaced by "Measurements" Done.

- Figure 3- The generalized ozone climatology shown as "a" gray band represents—— On average, all reanalysis products "underestimate" [O3]. Done.

- Line 172- Tromsø "where" [O3] Done.

- Line 188 - larger negative deviation from "observations" in DJF and MAM. Done.

- This indicates that CAMSRAQ might have different issues depending on the region of interest. "Different issues or different uncertainties- maybe this could be elaborated?" Thank you for pointing this out. We elaborate on the sentence and rephrase: *This indicates that underlying uncertainties in CAMSRAQ manifest differently at higher latitudes. Enhancements that lead to better model performances in mid latitudes, hence, do not necessarily affect results in the Arctic and subarctic in the same way.*

- Conclusions Line 248: You say "We confirm that a high spatial and temporal resolution, state-of-the-art mechanistic removal processes (land–atmosphere–ocean), and assimilation of in situ observations at ground-level are a must to constrain reanalysis products," but have you really confirmed it or explained that the land–atmosphere–ocean interaction is applied here? You are right. We have, in fact, not proven that in particular. What we confirm is that *the assimilation of vertically ozone profiles, if applicable ground-level observations, and a higher spatiotemporal resolution lead to better constrained reanalysis products.*

- Line 263 – "updates may "also" play a role" Done.

---

## Author Response (AR2)

**Authors' response**

To acp-2021-527 Editor decision: Publish subject to technical corrections (20 Sep 2021):
Thank you Jens-Uwe for your smooth handling of the review process and your useful comments. We really appreciated it.
We have addressed the issues as follows.

- You use the term "divergence" (fig 4, lines 194, 205) which is correct by language. As this term is also used in analysis you may think of replacing it by "deviation". But I leave this up to you. Thank you for pointing this out. We have replaced the term accordingly.

- You use [$O_3$] throughout the manuscript for the description of the ozone mixing ratio (in ppb). Normally, in the chemistry literature [X] refers to the concentration of a species X (in molecules per $cm^3$), not the mixing ratio. I should have noted that in the first round. It may be rather complicated to change this throughout the manuscript and figures (preferred), but at least I would ask you to explain this writing style once in the beginning. It is indeed difficult to replace all labels through out the manuscript at this stage. We add a short explanation in Section 1 after its first occurrence: "Here, [$O_3$] refers to the concentration as volume mixing ratio (VMR) of ozone in ppb."